# Effects of Milk Replacer-Based *Lactobacillus* on Growth and Gut Development of Yaks' Calves: a Gut Microbiome and Metabolic Study

Yaping Wang,[a] Miao An,[a] Zhao Zhang,[a] Wenqian Zhang,[a] Muhammad Fakhar-e-Alam Kulyar,[a] Mudassar Iqbal,[a,b] Yuanyuan He,[a] Feiran Li,[a] Tianwu An,[c] Huade Li,[c] Xiaolin Luo,[c] Shan Yang,[c] Jiakui Li[a,d]

[a]College of Veterinary Medicine, Huazhong Agricultural University, Wuhan, China
[b]Faculty of Veterinary and Animal Sciences, The Islamia University of Bahawalpur, Bahawalpur, Pakistan
[c]Sichuan Academy of Grassland Science, Chengdu, People's Republic of China
[d]College of Animals Husbandry and Veterinary Medicine, Tibet Agriculture and Animal Husbandry University, Linzhi, Tibet, People's Republic of China

**ABSTRACT** The gut microbiota and its metabolic activities are crucial for maintaining host homoeostasis and health, of which the role of probiotics has indeed been emphasized. The current study delves into the performance of probiotics as a beneficial managemental strategy, which further highlights their impact on growth performance, serologic investigation, gut microbiota, and metabolic profiling in yaks' calves. A field experiment was employed consisting of 2 by 3 factorial controls, including two development stages, namely, 21 and 42 days (about one and a half month), with three different feeding treatments. Results showed a positive impact of probiotic supplements on growth performance by approximately 3.16 kg ($P < 0.01$) compared with the blank control. Moreover, they had the potential to improve serum antioxidants and biochemical properties. We found that microorganisms that threaten health were enriched in the gut of the blank control with the depletion of beneficial bacteria, although all yaks were healthy. Additionally, the gut was colonized by a microbial succession that assembled into a more mature microbiome, driven by the probiotics strategy. The gut metabolic profiling was also changed significantly after the probiotic strategy, i.e., the concentrations of metabolites and the metabolic pattern, including enrichments in protein digestion and absorption, vitamin digestion and absorption, and biosynthesis of secondary metabolites. In summary, probiotics promoted gut microbiota/metabolites, developing precise interventions and achieving physiological benefits based on intestinal microecology. Hence, it is important to understand probiotic dietary changes to the gut microbiome, metabolome, and the host phenotype.

**IMPORTANCE** The host microbiome is a composite of the trillion microorganisms colonizing host bodies. It can be impacted by various factors, including diet, environmental conditions, and physical activities. The yaks' calves have a pre-existing imbalance in the intestinal microbiota with an inadequate feeding strategy, resulting in poor growth performance, diarrhea, and other intestinal diseases. Hence, targeting gut microbiota might provide a new effective feeding strategy for enhancing performance and maintaining a healthy intestinal environment. Based on the current findings, milk replacer-based *Lactobacillus* feeding may improve growth performance and health in yaks' calves.

**KEYWORDS** microbiome, metabolomics, *Lactobacillus*, gut development, yak calf, gut development

The yak (*Bos grunniens*) is an ancient ruminant with a mystique of the plateau (>3,000 m), thriving under extreme living conditions, such as low temperature and hypoxia (1). It has been connected intimately with the local human civilization and agriculture, providing basic survival resources, e.g., warm hides, dung for fuel, meat, and transportation (2). The

Address correspondence to Jiakui Li, lijk210@sina.com.

The authors declare no conflict of interest.

traditional grazing method relies mostly on natural pasture with limited supplementary feeding, so the yaks face deficient foraging resources on the plateau, especially during the calving season (3). In this context, neonatal calves are extremely susceptible to intestinal diseases caused by enteric bacterial imbalance. It results in inefficient digestion diarrhea, and poor growth, that further cause high mortality and morbidity (4). According to a number of recent studies, the mortality rate during early calves rearing remains remarkably high in most countries, e.g., 17% in Germany (5) and 5% in United States (6). Studies on milk or milk replacer (MR) feeding suggested that early feeding management is necessary to boost the immune system maturation, body development, and health of the calves (7). Additionally, when mother's milk and early feeding programs are implemented concurrently, the goal of better welfare during early period of rearing can be achieved easily (8). Hence, we hypothesized that early nutrition management might be important for the improvement of growth performance and well-being in yaks' calves.

The healthy nutritional management through probiotics is considered to be a vital feeding strategy for the prevention of neonatal gastrointestinal disturbance (9). Several studies have proved that probiotics effectively enhance immunity and feed efficiency and reduce diarrhea in neonate Holstein calves (10, 11). The proposed mechanism for direct-fed probiotics includes organic acids, digestive enzyme activities, immune system stimulation, and the production of various antagonistic factors, such as hydrogen peroxide, bacteriocins, and diacetyl, of which all have an obvious inhibitory effect against a wide range of pathogenic bacteria (12). The short-term therapy of *Lactobacillus reuteri* modulates intestinal microflora and improves proliferation, differentiation, immune defense, and barrier function formation in intestinal epithelial cells (13, 14). Yan et al. demonstrated that neonatal colonization of *Lactobacillus rhamnosus GG* improves the functional maturation of intestines and conferred lifelong health outcomes by enhancing the effectiveness against intestinal damage and inflammation (15). The potential role of certain *Lactobacillus rhamnosus GG* in regulating gut microbiota modulation and promoting intestinal metabolic profile is accomplished by the increase in beneficial microorganisms. So, the reduction of such microorganisms associated with health threats (16). Additionally, commensal bacteria in the gut systemically effect the molecules of microbial metabolism, such as short-chain fatty acids (17). These findings emphasized the need for identifying the role of certain *Lactobacillus* species that allow animals to develop a potential for intestinal maturation and growth. So, a comparable effect of *Lactobacillus* supplementation on stimulating neonatal intestine development is conceivable but requires further exploration in yaks' calves.

The development of the early intestinal microbiome acts as a bridge in the host-microbial relationship, which contributes to the maintenance of host health and normal physiological functions throughout the life. Ruminants have a massive and complex intestinal microbiome. Such an extraordinary symbiosis between a host and gut microorganism is linked to their mother's vaginal microbiome, which influences the development of their immune system and nutrition, as well as early colonization to form the microbiota's first colonizer community (18). The enormous potential for interactions between intestinal microorganisms indicates that early colonizers could affect the establishment of microbiota in long term (19). A recent study emphasizes the significance of intestinal microorganisms on the development and maturation of the neonatal immune system in the gut (20). Among those different microorganisms that colonize the host intestines, some entangle with the body's metabolism. For example, the potential metabolic capacities in the intestinal microbiome assist in absorbing the body's own inaccessible energy from diet and help in biotransformation to various xenobiotics (21, 22). The profiles of intestinal microbes may also alter the environment in the gut because of the metabolites. These microbes may be involved in immune system regulation and host-generated signaling (23, 24). Similarly, abundant metabolites that execute extensive metabolic activities in the gut are used efficiently by microbes for their own proliferation (25). As an example, tryptophan metabolites obtained from microbial sources might alter host physiology and behavior by decreasing the amount of tryptophan and activating the aryl hydrocarbon receptor by producing indole derivatives (26). In addition, short-chain fatty acids (SCFAs), produced

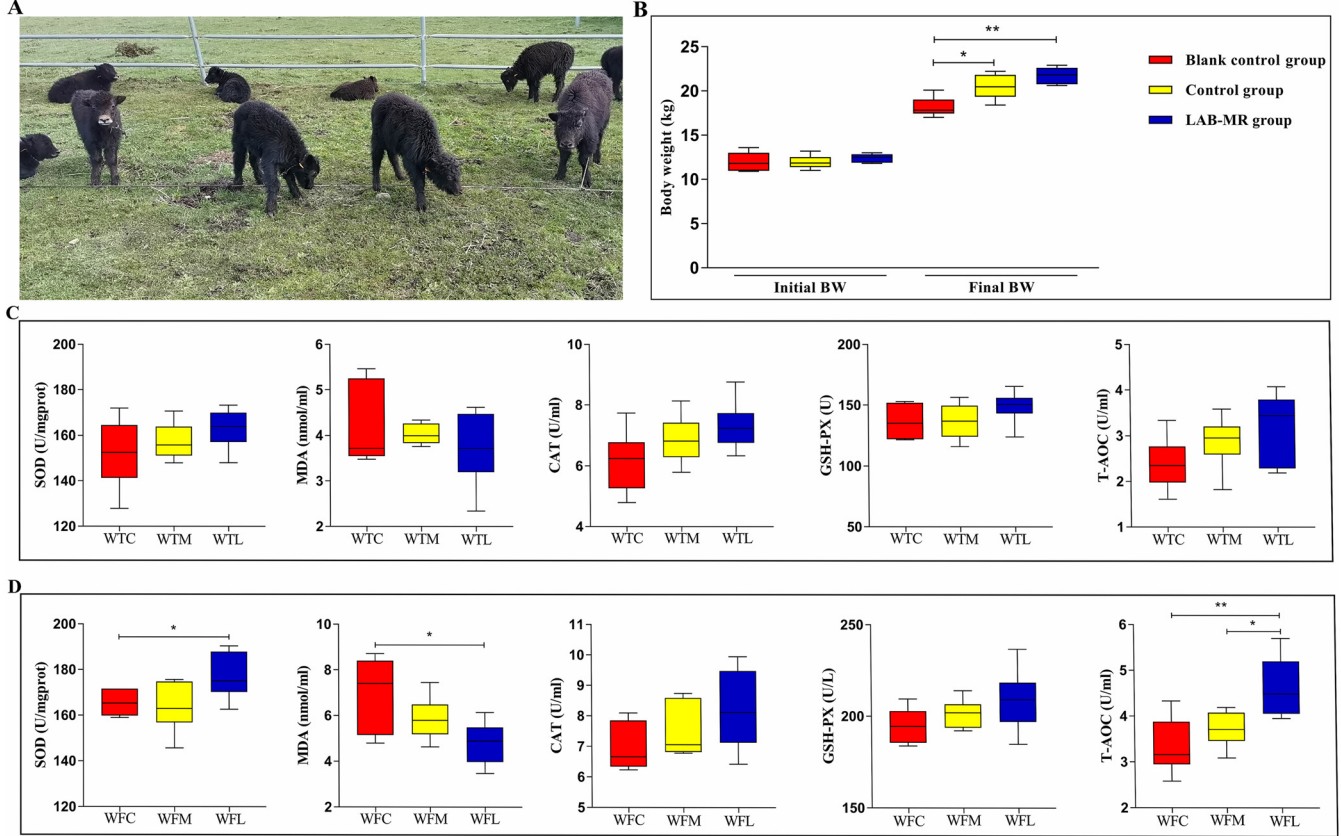

**FIG 1** Body weight and serum antioxidant properties of yaks' calves, supplemented with LAB in three different groups. (A) yaks' calves. (B) Initial and final body weight of calves. (C) Serum antioxidant properties of 21-day-old calves. (D) Serum antioxidant properties of 42-day-old calves. *, $P < 0.05$; **, $P < 0.01$.

by gut microorganisms during the digestion of dietary fiber, may also influence intestinal epithelial cells by modulating cell proliferation, gene expression, and immune response (27). Since the gut microbiota and metabolism have been connected to a wide range of disorders, it is not surprising that the intestinal microbiome and metabolism have been linked to a series of diseases (28, 29). The discovery of such relation between the gut microbiome and metabolome has promoted extensive research to characterize its taxonomic diversity (30). However, it is unclear how the composition and dynamics of microbiome and microbially mediated metabolic processes interact to maintain physiological benefits in calves. This limitation might be attributed to the variation in microbiome pathways and physiology adaptability between yak and other cattle species on plateau.

In this study, a controlled-field experiment was performed to investigate the growth, serum antioxidants, serum biochemical indicators, and gut development (i.e., bacterial communities and metabolites) under a milk replacer-based *Lactobacillus* (*Lactobacillus reuteri*). The study focused on 16S rRNA gene sequencing and untargeted liquid chromatography-mass spectrometry (LC-MS) metabolomics profiling to identify the dynamic distribution and interaction of the gut bacterial communities and metabolites in calves. The findings allowed us to develop a possible way for sustaining the health and well-being of yaks' calves.

## RESULTS

**Growth performance.** Fig. 1B presented values for initial and final body weight (BW). In our findings, the initial BW was not significantly different among treatment groups. Therefore, the difference in growth performance caused by the initial values was excluded. As expected, the final BW for calves receiving MR and milk replacer-based *Lactobacillus* (LAB-MR) tended to be higher than control calves ($P < 0.05$ or $P < 0.01$). Results showed a positive impact of LAB-MR on growth performance by approximately

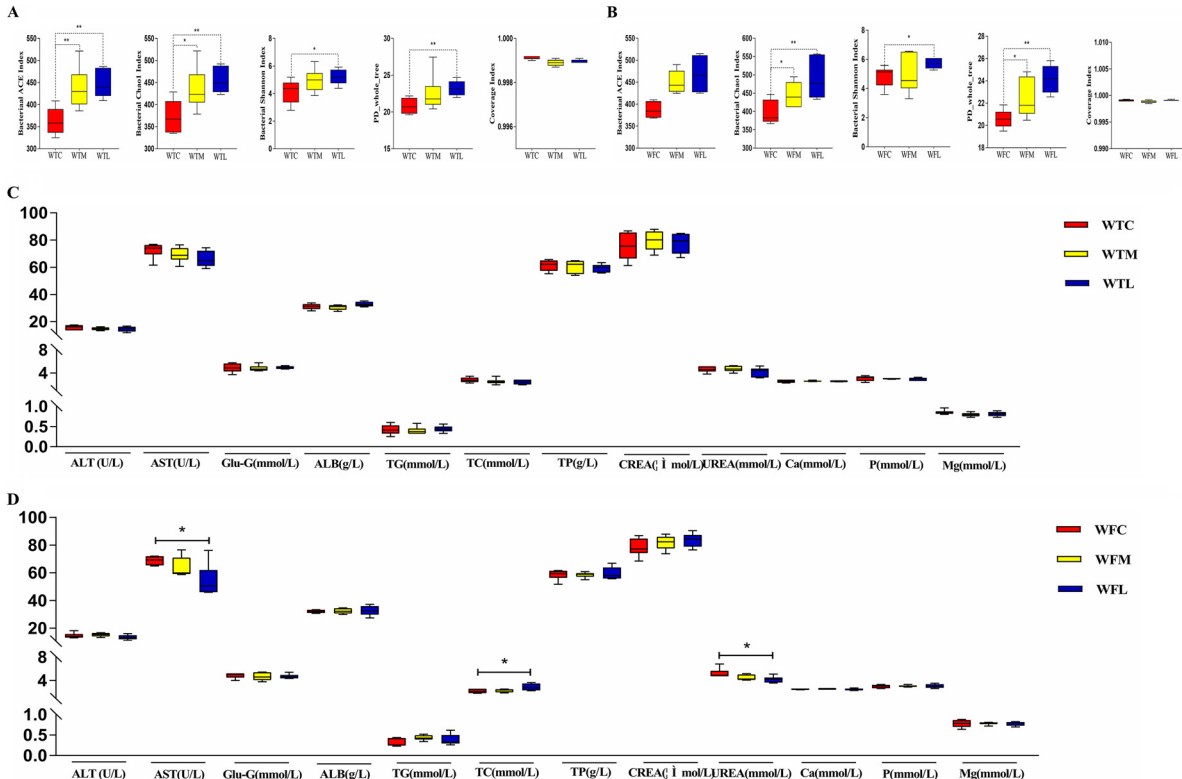

**FIG 2** Effects of LAB-MR supplementation on serum biochemical indexes and alpha diversity of gut microbiome in three different groups. (A) Alpha diversity of 21-day-old calves. (B) Alpha diversity of 42-day-old calves. (C) Serum biochemical indexes of 21-day-old calves. (D) Serum biochemical indexes of 42-day-old calves. *, $P < 0.05$; **, $P < 0.01$.

3.16 kg ($P < 0.01$) and 0.8 kg ($P > 0.05$) compared with blank control and control groups, respectively. Additionally, a management strategy for milk replacer-based *Lactobacillus* was found to be a potential tool for improving serum antioxidant properties (i.e., total antioxidant capacity [T-AOC], superoxide dismutase [SOD], and malondialdehyde [MDA]) (Fig. 1C and D) and serum biochemical parameters (i.e., aspartate aminotransferase [AST], total cholesterol [TC], and urea) ($P < 0.05$ or $P < 0.01$) (Fig. 2C and D).

**Feeding strategy increased the diversity of the gut microbiome.** We used amplicon sequencing to evaluate how the functional potential of the intestinal microbiome diversifies over the course of feeding strategy. Fecal samples from the calves at two different intervals (21 and 42 days) were collected, which passed strict quality control for raw sequence data (2,882,125) to obtain effective sequences (2,749,133). The average number per sample was 76,365 from bacterial populations. Multisample evidence for rarefaction curve, Shannon index, species accumulate curves, and coverage values demonstrated that almost all the bacterial populations were identified. Moreover, the current sequencing depth sufficiently covered the diversity of microbial communities (see Fig. S1A to C in the supplemental material). Based on 97% similarity, all of the V3/V4 regions of sequences were clustered into 2,166 operational taxonomic units (OTUs).

The multiple alpha diversity indices were measured to investigate the overall differences of microbial community richness and diversity. The intestinal diversity in the blank control was significantly lower than that of calves based on feeding management (Fig. 2A and B). The same condition was noticed for other diversity measurements (for Chao1, WTC, WTM, and WTL yielded 373, 435, and 455; and WFC, WFM, and WFL yielded 397, 445, 490; for ACE, WTC, WTM, and WTL yielded 362, 437, and 447; and WFC, WFM, and WFL yielded 387, 450, and 469; $P < 0.05$ or $P < 0.01$).

To assess the difference between intestinal microbial communities in the three groups, we calculated $\beta$-diversity. The unweighted pair-group method with arithmetic mean

(UPGMA) matrix distance and principal coordinates analysis (PCoA) clearly showed the variations in evolutionary information per sample (Fig. S1D and E). The similarity of each group showed a tendency of separation and partial overlap at the two different sampling points. The members in each group clustered in branches separately, suggesting that the early colonization of the gut microorganisms in juvenile animals is a nonfixed dynamically evolving structure. Additionally, the UPGMA matrix distance and PCoA clearly showed that the microbiota cluster in the blank control group was separated from the other groups (Fig. 3). The samples in control and LAB-MR groups clustered separately with the gradually expanding distance of the microbiome.

**Feeding strategy optimized the structure and composition of the gut microbiome in calves.** We evaluated the relative proportion of dominant bacteria in all samples at different taxonomical levels (Fig. S1F and G). Following the phylum level, *Firmicutes* was the most dominant phyla regardless of sampling time point and feeding management, which consisted of over 50.2% total sequences. Followed by *Bacteroidetes* in all groups (WTC, WTL, WTM yielded 18.02%, 34.75%, and 27.79%; WFC, WFL, and WFM yielded 17.04%, 35.25%, and 21.51%), except for the WTC group, the high abundance was replaced by *Proteobacteria* (20.83%). While at the genus level, *Faecalibacterium* and *Escherichia-Shigella* were notably enriched in WTC and *Bacteroides* in WTM, WTL, WFM, and WFL.

We next executed differential relative abundance analysis using Metastat to accurately explore the microorganisms among different groups (Fig. 4). The relative abundance of several intestinal communities between control groups showed a significant shift at phylum and genus levels, e.g., phyla *Fusobacteria*, *Tenericutes*, *Acidobacteria*, and *Deferribacteres* and genera *Bacillus*, *Barnesiella*, *Oscillospira*, *Clostridium_sensu_stricto_1*, *Anaerostipes*, and *Neisseria*. On the first sampling, the relative abundances of phylum *Patescibacteria* and genera *Bacteroides*, *Parabacteroides*, *Phascolarctobacterium*, *Blautia*, and *Anaerostipes* were significantly higher in the WTL group. The gut microbiome of the WTM group had a higher proportion of *Veillonella* and *Neisseria* than that of the WTL group. Moreover, the abundances of phylum *Bacteroidetes* and genera *Christensenellaceae_R-7_group*, *Lachnoclostridium_10*, *Clostridium_sensu_stricto_1*, *Bacteroides*, *Parabacteroides*, *Ruminococcaceae_UCG-004*, *Phascolarctobacterium*, *Blautia*, and *Anaerostipes* were significantly increased in WTL compared with those of WTC, while *Actinobacteria*, *Tyzzerella_4*, *Veillonella*, and *Neisseria* were significantly reduced. On the second sampling, the beneficial functional communities gradually enriched and microorganisms that threaten health reduced in the intestines of the LAB-MR group. In a brief view, the phyla *Bacteroidetes*, *Patescibacteria*, *Tenericutes*, and *Deferribacteres* and genera *Candidatus_Saccharimonas*, *Bacillus*, *Senegalimassilia*, *Odoribacter*, *Parabacteroides*, and *Anaerostipes* were more abundant in the LAB-MR group than those in control who were taking milk replacer. The phyla *Bacteroidetes* and *Patescibacteria* and genera *Candidatus_Saccharimonas*, *Alistipes*, *Lachnoclostridium_10*, *Defluviitaleaceae_UCG-011*, *Bacteroides*, *Ruminococcaceae_UCG-004*, *Saccharofermentans*, *Barnesiella*, *Faecalibacterium*, *Senegalimassilia*, and *Parabacteroides* were enriched in LAB-MR compared with those in the blank control group. Furthermore, the *[Clostridium]_innocuum_group* and *Porphyromonas* were more common in the blank control group. These findings conveyed a message that the principal component of microbiota remained separate and exhibited partial overlap between control and blank control groups. At the same time, milk replacer-based probiotic feeding motivated a different intestinal microbiome profile, which was dominated by a variety of functional beneficial bacteria.

**Effects of *Lactobacillus*-feeding strategy on the gut metabolic profiling.** Gut metabolite evaluation was performed in the positive and negative ion modes. A total of 668 and 927 different features were examined. All data from 18 samples, including quality-control (QC) samples were examined by PCA and orthogonal projections to latent structures discriminant analysis (OPLS-DA), following positive and negative mode ionization in order to characterize the overall metabolomic changes of different groups. As shown in PCA scatterplots, yak calves in the blank control and control were separated clearly from yaks in the LAB-MR group, regardless of ion modes or sampling (Fig. 5I to L). Additionally, a further OPLS-DA analysis showed that the three groups could be obviously separated according to their metabolic differences. Also, the data within the Hotelling T2 ellipse (with the class separation; R2Y) accounted for more than

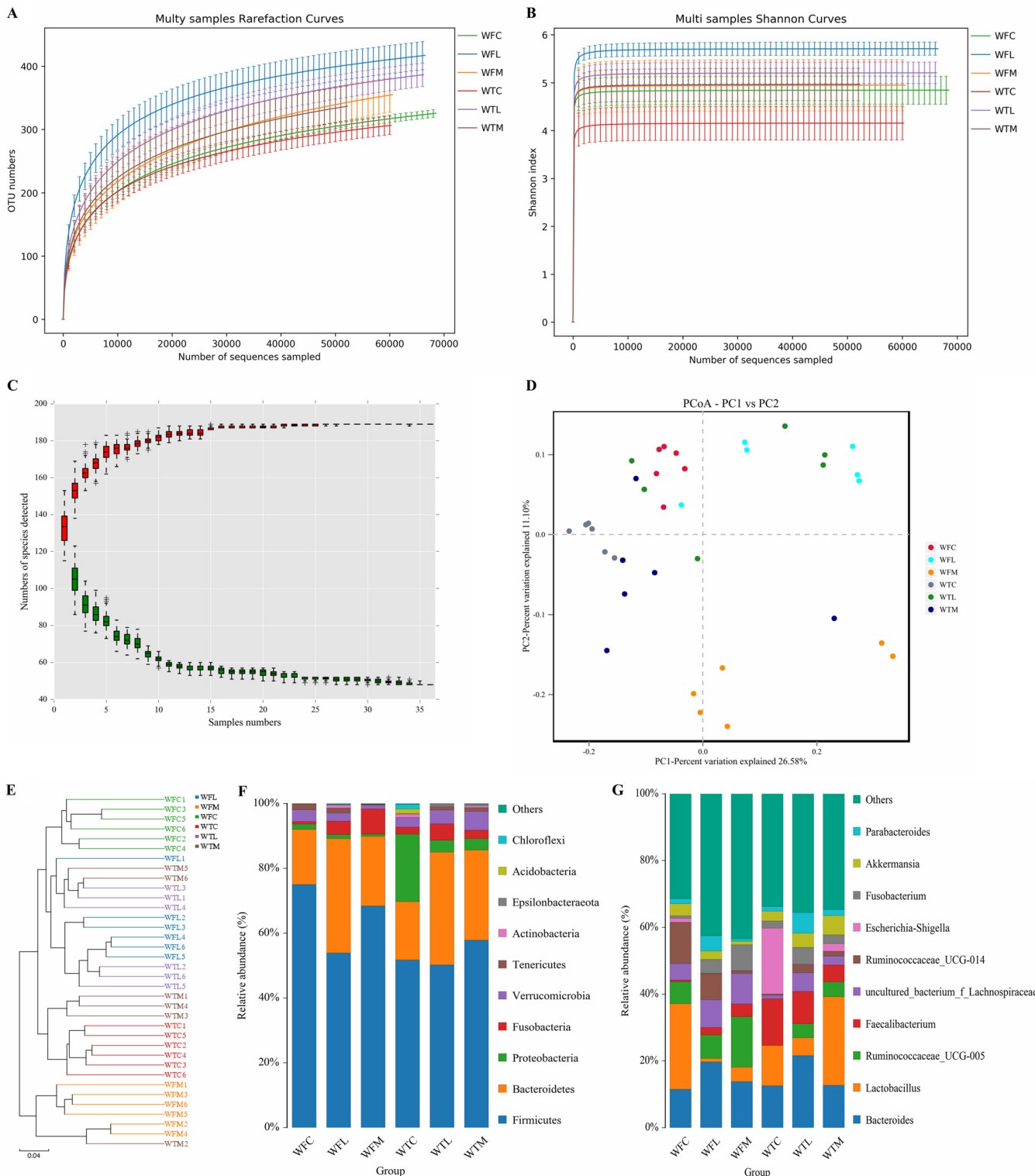

**FIG 3** Effects of LAB-MR supplementation on gut bacterial communities' structures. (A) Rarefaction curve; (B) rank abundance curve; (C) species accumulate curve; (D, E) represent the microbial similarity between groups by using PCoA scatter plot and UPGMA, respectively; (F, G) represent the analysis of microbial communities' structures at the phylum and genus levels, respectively.

0.991 (Fig. 5A to D). Meanwhile, the permutation test for OPLS-DA demonstrated that original points on the right were higher than R2 and Q2 values on the left. Also, the Q2 regression curve had a negative intercept, which shows the reliability and validity of the OPLS-DA model (Fig. 5E to H).

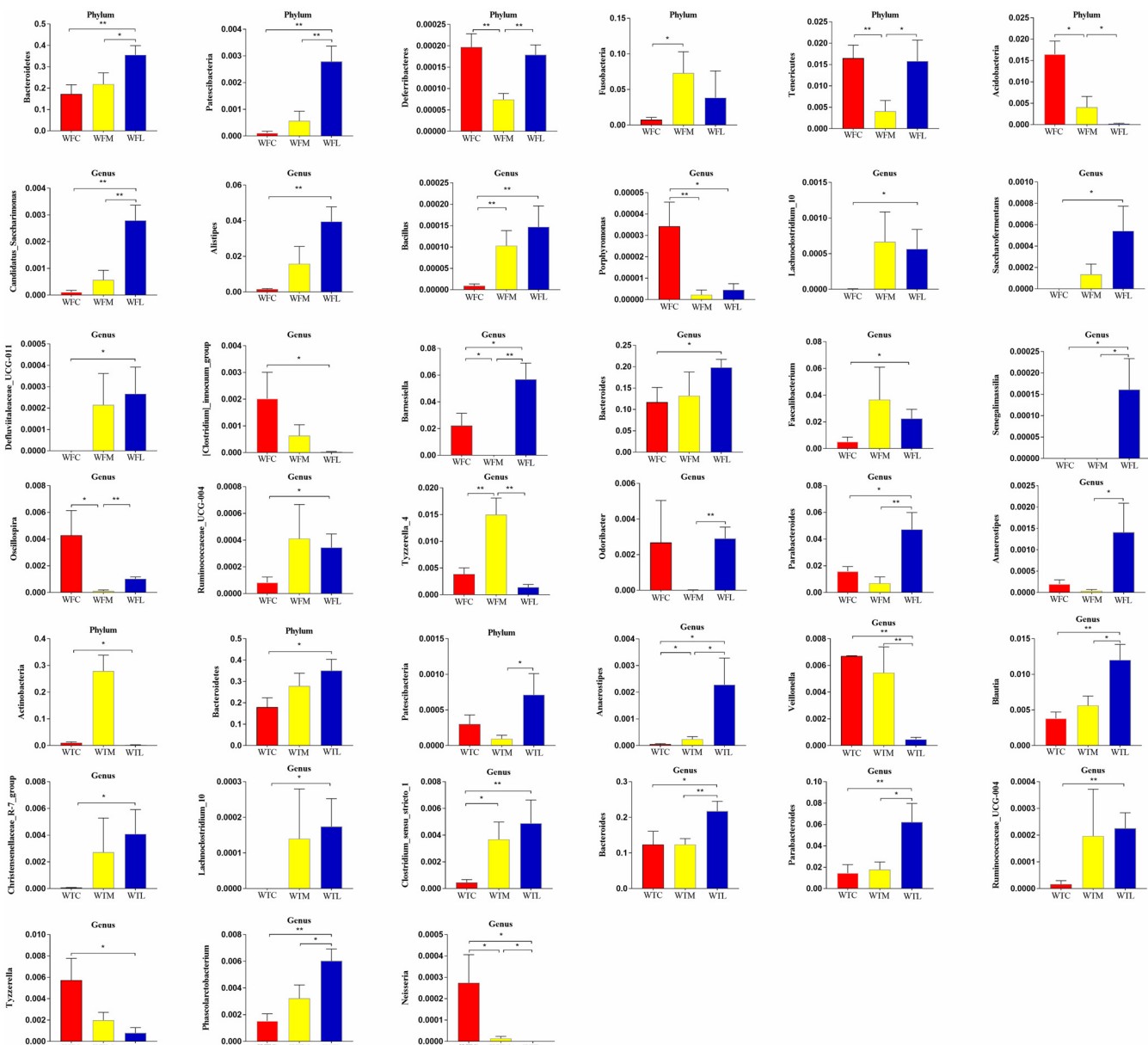

**FIG 4** Significant differences in the relative abundance of gut microbiota among three different groups at phylum and genus levels. *, $P < 0.05$; **, $P < 0.01$.

The gut metabolome was changed significantly after the feeding of LAB-MR. The differential metabolites between FC (i.e., WFC) and FW (i.e., WFL) (150 positively ionized metabolites and 224 negatively ionized metabolites) and differential metabolites between FM (i.e., WFM) and FW (185 positively ionized metabolites and 81 negatively ionized metabolites) were identified (see Table S1, Table S2, and Fig. S2A to D in the supplemental material). The 150 different metabolites between FC and FW could be classified mainly into 71 downregulated and 79 upregulated metabolites. Also, the 185 different metabolites between FM and FW were classified into 16 downregulated metabolites and 169 upregulated metabolites for the positive ionization analysis (Fig. S2A and B). The pathway enrichment analysis showed a total of 7 metabolic pathways that had a significant change ($P < 0.05$ or $P < 0.01$), including estrogen signaling pathway, GnRH secretion, steroid biosynthesis, cAMP signaling pathway, axon regeneration, thermogenesis, and shigellosis in the comparison between FC and FW. Vitamin digestion and absorption, glycerophospholipid metabolism, choline metabolism in cancer, novobiocin biosynthesis, alpha-linolenic acid metabolism, tropane metabolism, piperidine, and pyridine alkaloid biosynthesis were

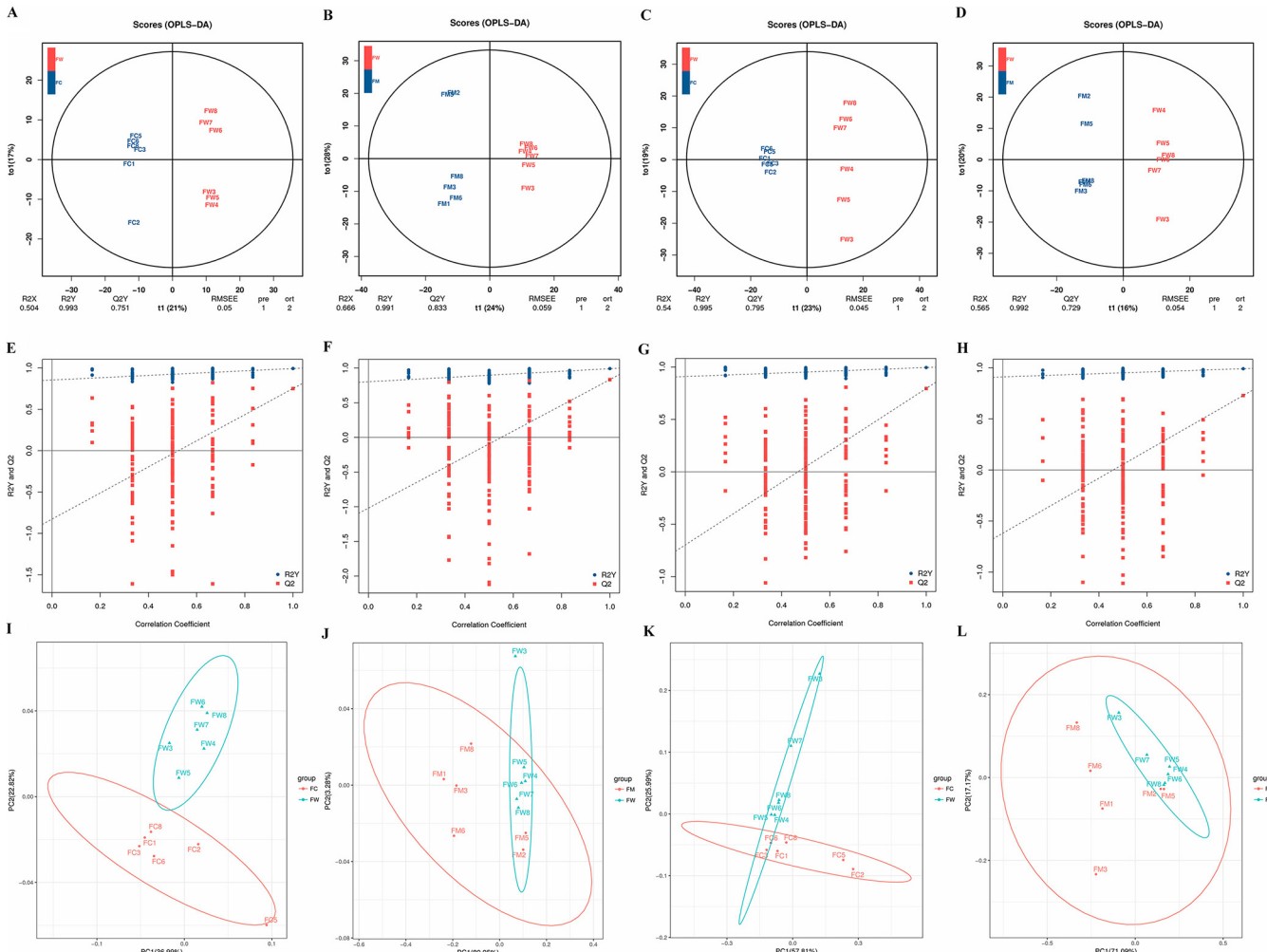

**FIG 5** Different metabolic patterns among three different groups. OPLS-DA score plot of gut metabolome in the ESI+ (A, B) and ES− modes (C, D). The permutation test for OPLS-DA in the ESI+ (E, F) and ES− modes (G, H) demonstrated the reliability of current OPLS-DA model. PCA score plot in the ESI+ (I, J) and ES− modes (K, L) assessed the metabolites' similarity between groups.

significantly changed between FM and FW (Fig. 6A and B). As shown in the volcano map in negative ionization, the 224 different metabolites between FC and FW could be classified mainly into 28 downregulated and 196 upregulated metabolites. A total of 77 upregulated and 4 downregulated metabolites were identified between FM and FW (Fig. S2C and D). Among different metabolic pathways ($P < 0.05$ or $P < 0.01$), five pathways with a significant impact value were pyrimidine metabolism, lipopolysaccharide biosynthesis, carbon fixation in photosynthetic organisms, ascorbate and aldarate metabolism, chlorocyclohexane, and chlorobenzene degradation between FC and FW. The other three pathways with significant impact were secondary bile acid biosynthesis, phenylpropanoid biosynthesis, and 2-oxocarboxylic acid metabolism between FM and FW (Fig. 6C and D).

**Correlation analysis between differential intestinal microbes and metabolites.** The multiomic nature of the data set identified the microbial characteristics and metabolites that were significantly different in their abundance after LAB-MR feeding. There might be a mechanism in place that connects metabolite concentrations to the abundance of microorganisms; however, this mechanism may be ameliorated by LAB-MR feeding strategies. For example, an increase in microorganism abundance may be accompanied by an increase in the concentrations of some by-products. To identify such a covariant relationship, we performed Spearman correlation analysis between altered fecal metabolites and perturbed intestinal microbes. The results showed that the increases in the genera *Candidatus_ Saccharimonas*, *Barnesiella*, *Odoribacter*, and *Parabacteroides* were positively correlated with

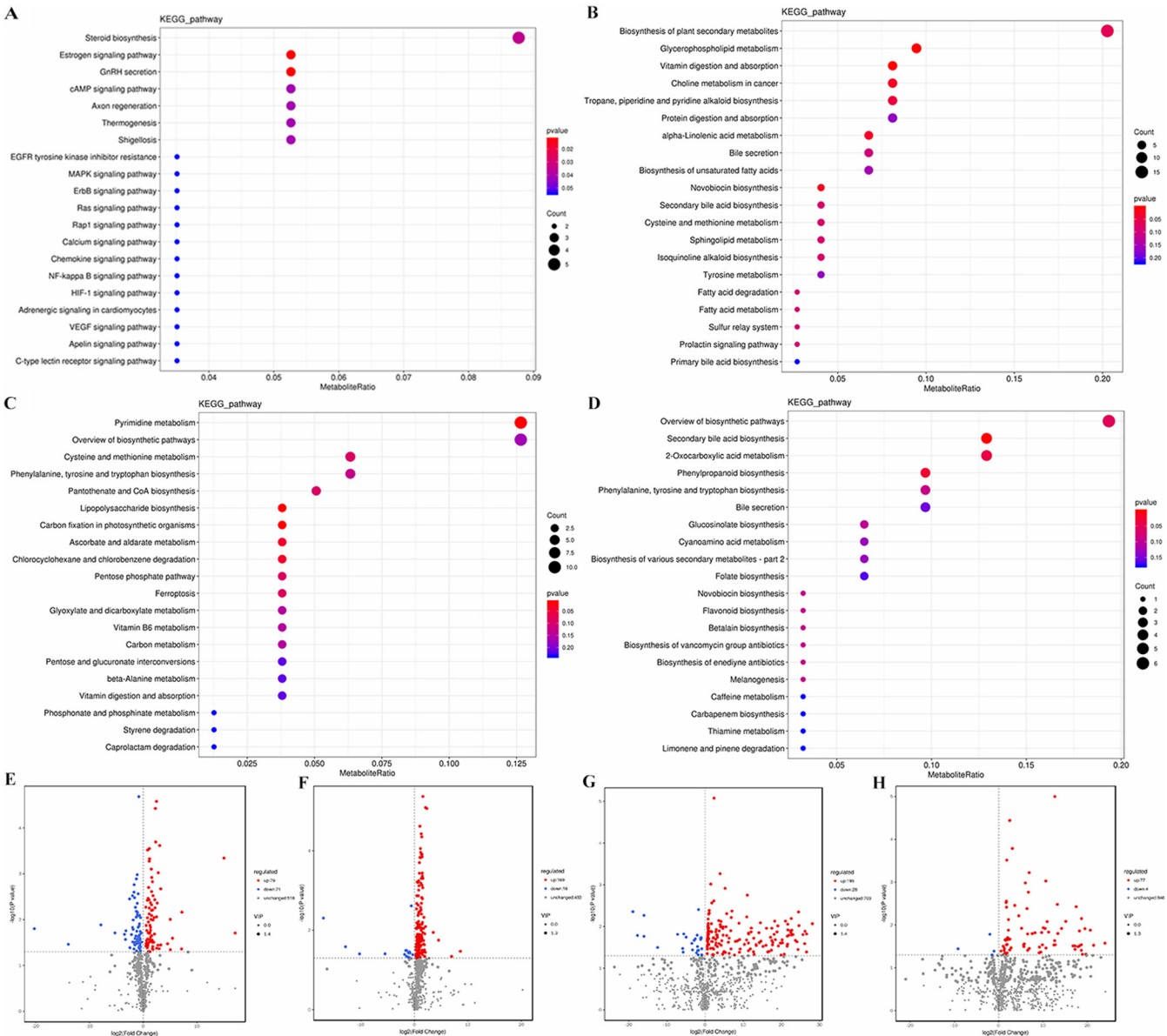

**FIG 6** Differential gut metabolites and metabolic pathways driven by LAB-MR supplementation. The metabolic pathways that were significantly changed in the ESI+ (A, B) and ES− modes (C, D). Volcano map showed the metabolites that were significantly up- or down- abundant driven by LAB-MR supplement in the ESI+ (E, F) and ES− modes (G, H).

the upregulated metabolites of L-isoleucine, L-proline, and L-tyrosine, whereas it was negatively correlated with the downregulated metabolites, e.g., phosphorylcholine and choline (Fig. 7). Another metabolite was included in *Tyzzerella_4* genera, which had an unfavorable association with the aforementioned metabolites. Therefore, the changes in intestinal metabolites might associate with the alteration of gut microbiota to LAB-MR feeding.

## DISCUSSION

The gut microbiome is home to trillions of microbes that play a vital role in maintaining host homoeostasis (31). Although microbiota colonization is important, its composition and structure induce metabolic variations that further may cause alterations in phenotypes (32). Such colonization of the intestinal microbiome is highly malleable and influenced by different factors, e.g., growth, environmental conditions, diet, and physical activities (1). The conjoint analysis of microbiome and metabolome has been recognized as the most promising method for assessing host-microbiome

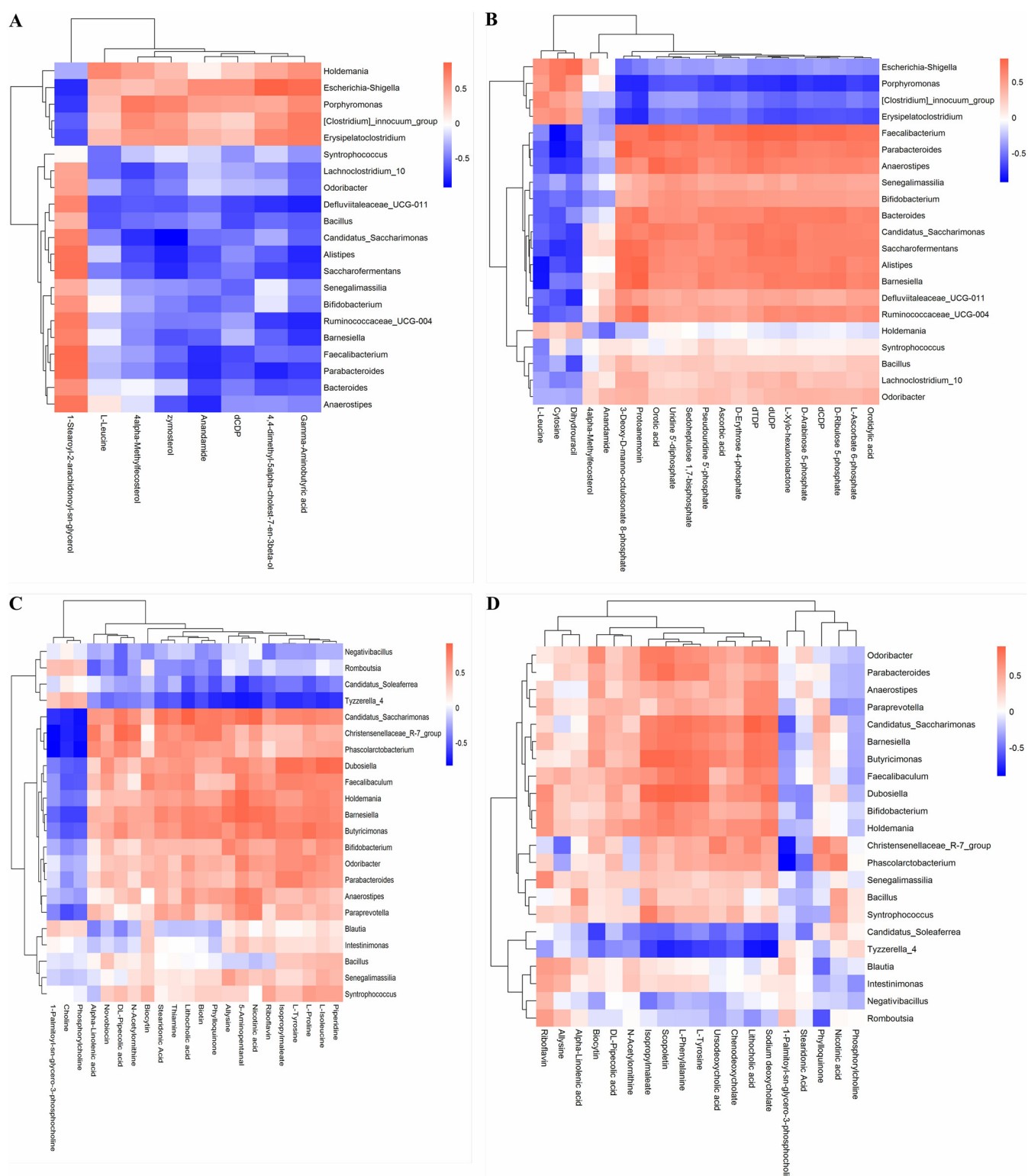

**FIG 7** Potential correlations between LAB-MR strategy-linked microbes and metabolites in the ESI1 and ES2 modes. (A, B) Represented covariation associations between FC and FW groups. (C, D) represented covariation associations between FM and FW groups.

interactions (33). However, few studies have used this method in yak calves. In the current study, we used a feeding strategy to examine the physiological effectiveness of *Lactobacillus* on the bacterial community and metabolic profile in yaks' calves. First, we confirmed the role of growth performance, serum antioxidant, and serum biochemical

indicators. Then we provided a detailed evaluation of the microbiome and metabolome mediated by *Lactobacillus*, which contributed to the identification of a possible feeding strategy for maintaining host health. The current study provided the first evidence that *Lactobacillus*-based milk replacer might be a potential feeding strategy for modulating the composition and relative abundance of intestinal microorganisms and metabolites. Also, we found multiple links between the gut microbiome (structure and taxonomic composition) and intestinal metabolites.

Supplementing *Lactobacillus* has been reported to improve feeding efficiency and body weight by modulating intestinal microflora, limiting pathogen invasion, and increasing villus height (34, 35). Similarly, Cox et al. demonstrated that probiotic promotes steady gut microbiota, stimulating digestibility and mucosal immunity (12). Timmerman et al. emphasized that *Lactobacillus* supplementation works best during early stressful periods. This hypothesis was confirmed by feeding five *Lactobacillus* and one *Enterococcus faecium* (36). In another study, probiotics were given to newborns and young animals, which were proven to induce some therapeutic effects in reducing the duration of intestinal disease (37). However, the effect of supplementing probiotics on the growth performance of the host seems uncertain. Szabo et al. reported that pigs supplemented with *Enterococcus faecium* had no difference in weight gain compared with negative-control pigs after challenge with *Salmonella enterica* serovar Typhimurium (38). These inconsistent responses might be caused by many potential factors, including pathogen loads, animal species, animal stress, and environmental factors (39). In our current findings, among three treatments groups, calves fed LAB-MR had more body weight than the blank control group ($P < 0.01$), whereas calves fed MR were intermediate among the three groups. It might be due to the crude protein (CP) content of the milk replacer that drove the young animals' frame growth (40). It could also be correlated with improving gastrointestinal tract development (41). This study provided the first evidence of improvement with *Lactobacillus*-based milk replacer in yak calves. Also, it further confirmed the possibility of LAB-MR as a feasible feeding strategy through serum antioxidant and serum biochemical indicators.

Animals develop and maintain a complex host-specific gut microbial community that includes three domains of life, i.e., *Archaea*, *Bacteria*, and *Eukarya* (19). The gastrointestinal tract of a newborn is colonized immediately by various microorganisms after birth. The process of such colonization has been identified as a coevolution, caused by the interaction between host and microbes (42, 43). This remarkable symbiosis, formed by a variety of environmental factors, began during direct contact with the maternal vaginal microbiome during birth (18). The gut microbiome is a community of microorganisms playing a vital role in most physiological processes (44), while complex microbiotas also have a role in providing fertile ground for noncommunicable diseases and infections. For example, the gut microbiome has been linked to a variety of ailments, such as inflammatory bowel disease and irritable bowel syndrome (45, 46), whereas others are scarcely thinkable, e.g., autism spectrum disorder and Parkinson's disease (47, 48). A pattern of abrupt dynamic changes in microbiota is correlated with interference, which demonstrates a deterministic mechanism throughout the development (49). These shifts might be explained by the developing immune system, diet, and the initial effects of microbial colonies. Furthermore, the nondigested nutrient contents are catabolized by gut microorganisms. As a result, metabolites are transported, absorbed, or excreted by highly dynamic metabolic pathways (50). Previous research has revealed that the characteristics and concentration of gut metabolites contribute to our understanding of the effect of metabolism on their host (51). Importantly, intestinal metabolites nourish the gut epithelial cells as well as adjusting downstream signaling pathways (52), which further acts as a bond between the gastrointestinal tract and host health. It implies that intestinal microflora dysbiosis probably is not the sole inducer for changing host physiology, as the gastrointestinal tract-derived metabolites also have significant systemic impact, e.g., immune system modulation. Therefore, it is important to investigate host gut microbiome-metabolite interactions within specific animal species and management practices. Previous research reported that the administration

of *Lactobacillus* to newborn calves increased weight gain and decreased diarrhea incidence. These efficiencies were greater in early-weaned calves than those in adult calves, indicating more effectiveness for intestinal communities (53).

Research on the gut microbiome has provided resources that connect intestinal microbiome interactions to health-related results. Indeed, in this cross-sectional cohort, we found that calves fed with LAB-MR had the highest alpha indices among three groups ($P < 0.05$ or $P < 0.01$), reflecting the richness and diversity of bacterial communities. Previous research found that the microbiome interindividual variation reduced with age except microbial diversity (54). Newborn or young animals are more susceptible to infections or intestinal diseases, resulting in high morbidity and mortality. But this condition eases gradually with age (55). So, we hypothesized an association between gut microbiota diversity and susceptibility after using *Lactobacillus* supplementation in order to overcome intestinal diseases. Moreover, six branches in the evolutionary tree were associated with feeding strategy, revealing an adaptability in calf intestinal microbiota that was linked closely to dietary shift resources. However, host demographics, individual temperament factors, and infectious diseases all had a certain degree of influence on microbiome structure (56). As a result, more clarification of the gut microbiome among the yak calves was needed to adequately investigate the role of *Lactobacillus* supplementation in intestinal microbiota acquisition.

The differences of specific microorganisms intuitively reflected the intrinsic connection between LAB-MR supplement and gut microbiota composition. Our study found that calves in the LAB-MR group had the most abundant phyla (*Bacteroidetes*) and genera (*Lachnoclostridium_10*, *Bacteroides*, *Parabacteroides*, and *Anaerostipes*), regardless of sampling time. For young ruminants, *Bacteroidetes* play an important role in degrading carbohydrates and proteins for facilitating gastrointestinal immune system (57). *Bacteroides* were identified as potential microorganisms to regulate the intestinal environment for immunomodulation and healthy homeostasis. At the same time, *Lachnoclostridium_10* has been reported to work in response to changes in gut luminal proteins (58, 59). The members of genus *Anaerostipes* ferment xylitol to produce butyrate, a nondigestible carbohydrate which plays an important role in improving barrier function, while *Parabacteroides* is a producer of short-chain fatty acids (60, 61). As mentioned previously, a series of beneficial microorganisms are positively involved in regulating intestinal function and the immune system and reducing susceptibility to intestinal diseases (62). It conveys a message that the intestinal environment, which is more vulnerable to disease, drives the reduction of beneficial microbiota or the reduction in beneficial microorganisms. These findings might be the consequences of the host's gut microbiome toward a better structure, benefited by *Lactobacillus* supplementation. Furthermore, the largest percentage of *Actinobacteria* was detected in the blank control group, which may have been transformed readily into pathogenic bacteria when synergy with one partner or host was seen (63). In addition, *[Clostridium]_innocuum_group*, *Tyzzerella_4*, *Porphyromonas*, *Veillonella*, and *Neisseria* were enriched in the blank control of calves. These bacteria participate in gut bacterial dysbiosis and diseases propagation through opportunistic pathogens (64–68). Meanwhile, the functional microbes that maintain intestinal health or produce short-chain fatty acids were less abundant in the WFC group, i.e., *Candidatus_Saccharimonas*, *Alistipes*, *Lachnoclostridium_10*, *Defluviitaleaceae_UCG-011*, *Bacteroides*, *Ruminococcaceae_UCG-004*, *Bacillus*, *Saccharofermentans*, *Barnesiella*, *Faecalibacterium*, *Senegalimassilia*, and *Parabacteroides*, than those in the WFL group (69–71). In animal husbandry, gastrointestinal dysfunction and even diarrhea in juvenile ruminants are frequent, which negatively affects growth performance and even leads to the death. According to Steele et al., several bacteria have been alternating between weak and dominating communities, resulting in gut diseases (72). As a result, we hypothesized some inevitable associations between the immature gut microbiome and the susceptibility to intestinal diseases. This approach might explain why juvenile ruminants were more vulnerable than adults. Altered abundances of some microorganisms contributed to the unique microbial outcomes in LAB-MR-fed calves. Most prominently, the microbes associated with digestion and absorption

of protein or saccharolytic (*Bacteroidetes, Senegalimassilia*), intestinal environment health (*Deferribacteres, Patescibacteria, Tenericutes, Candidatus_Saccharimonas, Barnesiella, Bacteroides, Phascolarctobacterium,* and *Blautia*), and SCFAs producers (*Oscillospira, Odoribacter, Parabacteroides,* and *Anaerostipes*) were noticed (73, 74). A recent study supported the potential use of probiotics in routine husbandry practices (75).

The most common varieties of microorganisms that are incorporated into husbandry include *Lactobacillus* spp., *Bifidobacterium* spp., *Streptococcus* spp., *Bacillus* spp., and *Enterococcus* spp. Normally, the prophylactic application is based on the conceded mechanism of *Lactobacillus* with the restoration or establishment ability of a beneficial microbiota in the gastrointestinal tract of young animals. A meta-analysis investigated the effect of probiotic administration on the fecal microbiota and health of calves. According to the results, the microorganisms that threaten health were reduced (76). These reports support our findings in this study, suggesting that shifts in these functional gut bacteria are related to the better feeding management (LAB-MR). Moreover, the dynamic distribution, interaction of the gut microbiome, and metabolites are still needed to be evaluated under the effect of *Lactobacillus* administration in calves.

Furthermore, the impact LAB-MR supplementation on the gut metabolome was assessed using an untargeted metabolomics technique, i.e., LC-MS. Metabolites drive the crucial cellular functions, for example, energy production and storage, which affects gut conditions. Metabolomics can detect subtle alterations in biological pathways because of its inherent sensitivity to investigate the potential mechanism of various physiological conditions. We performed an enrichment analysis to evaluate broad classes of metabolites that were significantly up- or downregulated. In total, a considerable number of metabolites were changed due to LAB-MR. The pathway enrichment analysis showed that metabolites, including pyrimidine metabolism, lipopolysaccharide biosynthesis, carbon fixation in photosynthetic organisms, ascorbate and aldarate metabolism, chlorocyclohexane and chlorobenzene degradation, estrogen signaling pathway, GnRH secretion, steroid biosynthesis, cAMP signaling pathway, axon regeneration, thermogenesis, and shigellosis were altered significantly between FC and FW. Finally, significant differential metabolites were identified by overlapping the significant metabolites and annotated metabolites in enriched pathways ($P < 0.05$; variable influence on projection [VIP], $>1$). Here, the ascorbic acid and orotic acid (OA) were significantly high in FW, whereas dihydrouracil and 4,4-dimethyl-5alpha-cholest-7-en-3beta-ol had the opposite trend. Since the discovery of ascorbic acid (vitamin C) in 1920s, no other chemical has been proven to have an extraordinary effect (77). It is an antioxidant, which efficiently scavenges reactive oxygen species (ROS) and toxic free radicals. OA is another essential and versatile molecule for regulating genes that involve the development of host cells and tissues. Early nutrition research identified it as vitamin B13, which is a precursor in the biosynthesis of pyrimidines (78). Gastrointestinal toxicity has been attributed to dihydropyrimidine dehydrogenase (DPD) deficiency when fluorouracil is administered to an absolute and partial DPD deficiency population (79), while 4-alpha-methyl-5-alpha-cholest-7-en-3-beta-ol has been reported to induce oxidative stress to the host (80). It is well known that young yaks can easily face gastrointestinal disorders due to harsh living conditions (cold and hypoxia), stress response, unhygienic forage, or weather mutation (81). We speculated that the metabolites that endanger host health can disturb intestinal microflora, which further increases their vulnerability to intestinal diseases. Moreover, we found that LAB-MR supplementation significantly increased the concentration of several metabolites associated vitamins, e.g., thiamine, riboflavin, and phylloquinone. Thiamine (vitamin $B_1$) is synthesized by plants, fungi, and microorganisms. Its derivatives have a nonenzymatic role in regulating stress response and signal transduction pathways, associating with harsh environmental factors (82). Riboflavin belongs to the B vitamin family, and its intake probably has protective effects on a series of medical conditions, such as ischemia, while these biological effects have been investigated widely for their anti-inflammatory, antioxidant, and antinociceptive properties (83). Phylloquinone (vitamin K1) plays a necessary role in bone and vascular metabolism (84), while various clinical abnormalities occur under the conditions of biotin (vitamin H) deficiency, which include growth retardation and dermatological abnormalities (85). In the FW group, we observed an increased level of metabolites associated with

amino acids, i.e., L-isoleucine, L-proline, L-tyrosine, and L-phenylalanine, of which all have specific functions that make them beneficial to the host. In the gastrointestinal tract, amino acids, which are necessary precursors for synthesizing proteins and polypeptides, have been identified as markers of protein metabolism (86). In the healthy gut, chenodeoxycholates are decomposed to secondary bile acids, contributing to the digestion of lipids (87), while the ursodeoxycholic acid (formed from the transformation of deoxycholic acid) contributes to regulating lipid metabolism and intestinal barrier integrity (88). Several molecular classes, e.g., nicotinic acid, novobiocin, and sodium deoxycholate, are associated with moderating intestinal health. These classes inhibit pathogenic growth that significantly depletes in FM compared with FW (89, 90). The gut microbiome interacts with the host in a variety of ways, of which one is via a range of metabolites (created as end or intermediate products of microbial metabolism). The integrative analysis described the potential correlation between gut microorganisms and metabolites, driven by LAB-MR feeding strategy. The changes and potential relations, along with the previously mentioned beneficial amino acids and vitamins, are consistent with previously suggested host gut homoeostasis and health from the prospective of potential probiotic properties of *Lactobacillus* supplementation (36).

**Conclusion.** Overall, the changes in final body weight suggested milk replacer-based *Lactobacillus* had a positive impact on growth performance. The feeding in yaks' calves had the potential for improving serum antioxidant properties and serum biochemical parameters. The bacterial community of LAB-MR calves exhibited higher diversity and richer symbiotic microorganisms than that of the control groups. Additionally, the gut was colonized by a succession of microbiota that assembled into a more mature microbiome driven by LAB-MR. The gut metabolic profiling was also significantly improved after LAB-MR, i.e., the concentrations of metabolites and the metabolic pattern. The current study involved a conjunction of gut metabolomics and bacterial community analyses between functional bacteria and metabolites in yaks' calves that were significantly prompted by a milk replacer-based *Lactobacillus* feeding strategy. Such integrative information contributed to the development of efficient, healthy, and ethical modern husbandry strategies for yaks' industry.

## MATERIALS AND METHODS

**Field methods.** This experiment was conducted at Maiwa Yak Breeding Base in Aba Tibetan and Qiang Autonomous Prefecture (>3,500-m altitude), in the eastern part of the Qinghai-Tibet Plateau Sichuan, China. All calves were born and enrolled at the experimental unit on the same day. After visual inspection, 18 competent calves (0 d, born within 24 h) with similar health status and immunization were recruited.

Calves were assigned randomly for further treatments by a veterinary technician, who was unaware of the experiment. Three groups were allocated with 6 calves per treatment (half male and half female) at $3 \pm 1$ day of age. Treatments were as follows: (i) control group (WTM and WFM), mother's milk + milk replacer (MR) at 0.5 L/day MR per calf (06:00 h and 18:00 h); (ii) probiotics group (WTL and WFL), mother's milk + milk replacer-based *Lactobacillus* (LAB-MR) at 10 g/day LAB product isovolumetric to MR (06:00 h and 18:00 h); and (iii) blank control group (WTC and WFC), mother's milk. The LAB product was a water-soluble powder that provided $1 \times 10^8$ CFU/g of *Lactobacillus reuteri*. The potential probiotic properties of this strain had been described according to our previously published study (14). The nonmedicated MR containing 20% CP, 17% fat, and 10% ash content was purchased from Chengdu Xingguang Quan Nutrition Food Co., Ltd. with producer no. Q/91510115794900448U.3-2020. The MR was reconstituted with warm water (46°C) to attain a solid concentration of 10%. Before animals were fed, the incubator was used at 37°C to ensure that MR or LAB-MR were homogeneous. Upon completion of supplement consumption, the bucket was rinsed and kept sterile every day. Numbered necklaces instead of ear tags were assigned to calves for providing natural habits of the calves before letting them graze with their dams. Other than the experimenter, the entries were limited to two herders and a staff veterinarian. The job of the two herders was to find the calves with necklaces twice/day (morning and afternoon) to execute the feeding protocol and then lift all the calves back into the herd. Calves were assumed to have fed from the same mother, which was an uncontrollable factor. Moreover, there is no record of calves' grass intake, as it was impossible to track in the natural pasture. Furthermore, all the yaks drank water from the ranch's mountain spring.

**Body weight, rectal temperature, and blood sample collection.** The body weight of all calves was measured before the morning feeding. Rectal temperature was measured four times per week (Monday, Wednesday, Friday, and Sunday) by using a handheld thermometer before morning feeding. Blood samples for hematology analysis were collected two times during the experiment (day 21 and day 42) by using a 21-gauge needle via jugular venipuncture into no-additive evacuated tubes. The centrifugation of all samples at $3,000 \times$ rpm/min was performed just after 30 min at room temperature, and then serum samples were stored at $-80°C$ in quadruplicates until hematology analysis. The analyzed indicators were alanine aminotransferase (ALT), aspartate aminotransferase (AST), glucose (Glu-G), albumin (ALB),

triglycerides (TG), total cholesterol (TC), total protein (TP), creatinine (CREA-S), urea, calcium (Ca), phosphorus (P), and magnesium (Mg).

**Fresh fecal samples collection.** Fresh fecal samples were collected on day 21 (named blank control; control; and LAB-MR groups WTC, WTM, and WTL) and day 42 (named blank control; control; and LAB-MR groups WFC, WFM, and WFL) from each calf's rectum via digital stimulation or fresh defecation. Then samples were placed immediately in liquid nitrogen and shipped on dry ice to the laboratory. We performed metagenomics profiling and untargeted metabolomics on three cohorts. The fecal samples were stored at −80°C prior to high-throughput sequencing or metabolomics profiling as described below.

**Microbiome sample processing and sequencing.** Total genomic DNA in fecal samples collected at days 21 and 42 was extracted using the QIAamp fast DNA stool minikit (Qiagen, Inc.), according to the manufacturer's instructions. Then, a 1% agarose gel was used to monitored genomic DNA concentration and purity. To perform the subsequent pyrosequencing, the V3-V4 region of bacterial 16S rRNA gene was amplified using universal primers (338F, 5′-ACTCCTACGGGAGGCAGCA-3′; and 806R, 5′-GGACTACHVGGGTWTCTAAT-3′). The PCR amplification was executed at following conditions: the annealing temperature was 57°C during the 25 PCR cycles. The PCR products were confirmed by 1% gel electrophoresis, cleaned, and normalized using the SequalPrep normalization plate kit (Life-Technologies, CA). According to the manufacturer's instructions, purified amplification PCR productions were generated in a sequencing library using the Next Ultra DNA library prep kit (New England BioLabs [NEB], USA). After quality inspection using a bioanalyzer (Agilent Technologies, USA) and quantitative PCR (qPCR), only libraries with a single peak and concentration of more than 2 nM were kept and used for high-throughput sequencing. The qualified library was sequenced on the HiSeq 6000 platform (Illumina, San Diego, CA), targeting the sequences with paired-end reads.

**Bacterial metagenome bioinformatics and statistical analysis.** The raw reads from high-throughput sequencing were quality filtered to obtain effective reads through performing the following preprocedures. Primer sequences were trimmed to obtain clean reads by using cutadapt 1.9.1 software, and then UCHIME v4.2 software was used to identify and remove chimera sequences. Furthermore, the effective sequence alignment performed with the SILVA database was clustered into operational taxonomic units (OTUs) using Quantitative Insights into Microbial Ecology (QIIME) software (Uparse v7.0.1001), with a threshold of ≥97% sequence similarity. The taxonomic assignment of all sequences and microbial composition were analyzed based on normalized output data. The calculations of alpha diversity (Chao1, ACE, Shannon, and PD_whole_tree) were performed in QIIME software. The rarefaction curve, Shannon curve, rank abundance curve, species accumulate curve, and Good's coverage were visualized in R software (v3.6.0) to demonstrate species abundance and evenness and reflect whether the current sequencing depth covered the vast majority of species information. Principal coordinates analysis (PCoA) and unweighted pair-group method with arithmetic mean (UPGMA) investigated the similarities between groups or individuals. Metastats analyses (performed *t* test) were conducted at different taxonomic levels (phylum and genus) for assessing the difference in the relative abundance of intestinal microbiome members for finding biomarkers between groups of samples.

**Stool sample processing and metabolite profiling analysis.** Fecal sample metabolomics profiles were performed using liquid chromatography-tandem mass spectrometry (LC-MS/MS). Chromatographic analysis was executed on the Waters Acquity ultra-high-performance system (1290 UHPLC; Agilent), and high-resolution mass spectrum (HRMS; TripleTOF 5600; AB Sciex) enabled the nontargeted measurement of metabolites. Briefly, stool samples (about 50 mg) were triturated in precooled methanol (CNW Technologies) by using a bead mill (TissueLyser; Qiagen). Next, the mixtures were incubated successively at 0°C for 10 min and −20°C for 1 h and then were centrifuged at 13,000 rpm at 4°C for 15 min. The sample supernatant (20 $\mu$L) was collected and mixed together to 200 $\mu$L as a quality control (QC) sample and then was injected into the LC-MS/MS system for further analysis.

The mass spectrometry data were obtained from the AB 5600 TripleTOF mass spectrometer under the control of Analyst software (Analyst TF 1.7; AB Sciex). Time-of-flight (TOF) parameters were set as follows: bombardment energy, 30 eV. The electrospray ionization (ESI) ion source parameters are set as follows: atomization pressure (GS1), 60 Psi; auxiliary pressure, 60Psi; air curtain pressure, 35 Psi; temperature, 650°C; and spray voltage, 5,000 V (positive ion mode) or −4,000 V (negative ion mode). Raw LC-MS data were proceeded using Genedata Expressionist software (v9.0) to remove chemical noise. The procedure included chromatographic peak detection, integration, normalization, and alignment retention times between samples. The processed data were then used to execute principal-component analysis (PCA) and orthogonal to partial least-squares discriminate analysis (OPLS-DA) for visualizing the metabolic profiling differences between groups. Taken together, the significantly different metabolites were considered regarding variable influence on projection (VIP) of >1 and a *P* value of <0.05.

**Statistical analysis.** The SPSS (v21.0) software was used to perform statistical analyses. GraphPad Prism (v7.0) software was used to draw box plots. The values were presented as mean ± standard (SD). Statistical significance was identified as a *P* value of <0.05.

**Ethics approval.** The animal-specific procedures were approved by the animal ethical committee of Huazhong Agricultural University. The experiments did not involve any invasive operations on animals.

**Data availability.** The raw sequence reads were submitted to the NCBI public database (SRA) with accession no. PRJNA818126.

## SUPPLEMENTAL MATERIAL

Supplemental material is available online only.
**SUPPLEMENTAL FILE 1**, PDF file, 0.1 MB.

## ACKNOWLEDGMENTS

Y.W. and J.L. provided the research idea. M.A., Z.Z., W.Z., Y.H., F.L., H.L., T.A., X.L., and S.Y. contributed animals, reagents, materials, and analysis tools. Y.W. wrote the manuscript. M.F.K. and M.I. revised the manuscript. All authors read and agreed to publish the final manuscript.

The current study was supported by Chinese Agricultural Research Systems (CARS-37), Tibet Autonomous Region Science and Technology Program (XZ202001ZY0044N).

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
