## [Reviewer comments · Microbiology Spectrum]

Microbiology Spectrum

Effects of milk replacer–based *Lactobacillus* on growth and gut development of yak calves; A gut microbiome and metabolic base study

Yaping Wang, Miao An, Zhao Zhang, Wenqian Zhang, Muhammad Fakhar-e-Alam Kulyar, Mudassar Iqbal, Yuanyuan He, Feiran Li, Tianwu An, Huade Li, Xiaolin Luo, Shan Yang, and Jiakui Li

Corresponding Author(s): Jiakui Li, College of Veterinary Medicine, Huazhong Agricultural University, Wuhan, 430070, PR China

Review Timeline:

Submission Date:	March 29, 2022
Editorial Decision:	May 3, 2022
Revision Received:	May 16, 2022
Accepted:	June 8, 2022

Editor: John Chaston

Reviewer(s): The reviewers have opted to remain anonymous.

Transaction Report:

DOI: <https://doi.org/10.1128/spectrum.01155-22>

Prof. Jiakui Li
College of Veterinary Medicine, Huazhong Agricultural University, Wuhan, 430070, PR China
Shizishan Road 1
Wuhan
China

Re: Spectrum01155-22 (Effects of milk replacer-based Lactobacillus on growth and gut development of yak's calves; A gut microbiome and metabolic base study)

Dear Prof. Jiakui Li:

I have received the reviews of your manuscript entitled "Effects of milk replacer-based Lactobacillus on growth and gut development of yak's calves; A gut microbiome and metabolic base study". The reviewers were generally positive about the outlook and theme of the work but found that it was not possible to properly evaluate the science because the English in the manuscript needed to be improved. Therefore, even though Spectrum does not normally consider resubmissions of rejected manuscripts, we are willing in this instance to make an exception. However, I would like to note that I will be looking for a robust revision that makes the results of the manuscript able to be clearly interpreted. To improve the English you may wish to work with a colleague or to use one of the services recommended here <https://journals.asm.org/content/language-editing-services>. Also, please note that the revision you submit will be subjected to another round of review and I will be looking for you to address Reviewer 1's comments in your revision as well. Finally, if you choose to pursue this option, please write to the editorial spectrum@asmusa.org BEFORE submitting your revised paper.

I am sorry to convey a negative decision on this occasion, but I hope that the enclosed reviews are useful. Please note, rejections from Microbiology Spectrum are final and your manuscript will not be considered by other ASM journals. We wish you well in publishing this report in another journal and hope that you will consider Spectrum in the future.

Sincerely,

John Chaston
Editor, Microbiology Spectrum

Reviewer comments:

Reviewer #1 (Comments for the Author):

Overall, the manuscript is interesting, the methods are appropriate, and the results are well presented. The manuscript could use an additional spelling and grammar check, and I included some minor corrections here:

line 14: correct to "highlights"

line 26" remove the hyphen

line 29: correct to "probiotics dietary supplements changes the"

line 33: correct to "collaterally"

line 39, 454, 458, 460, Fig 1 legend: correct to "yak calves"

line 42: italicize the species' name

line 60: correct to "probiotics is suspected to be a vital feeding strategy as"

line 62: while this sentence is true, since this paper focuses on cattle it would be more informative to include a reference on improving calf health here instead of one on human infants.

lines 68- 76: was this research on any species of Lactobacillus? I think many of the benefits are species or strain specific, so the authors should be more specific. For example, "...therapy of certain Lactobacillus species modulated..."

line 85: "establishment of microbiota"

line 103: "yak calves' microbiome"

line 108: I would change 'microbiome' to 'bacterial community' here and elsewhere in the manuscript because 16S only gives a fraction of the microbiome and is misleading to use.

line 136: correct to "diversity"

lines 158 - 159: italicize the genera names here

lines 185: I am not sure what the authors means by "kept tendency", please revise

line 193, 248, 250, 253, 323: revise to "yak calves"

line 404: "vitamins"

Figures: the text in the figures was too small to read since there were so many panels. I recommend moving some of these to supplementary, or splitting figures into two as needed to make the panels and text larger.

Reviewer #2 (Comments for the Author):

The manuscript submitted by Li et al. demonstrates a milk replacer-based lactobacillus in the yak's calves could increase the body weight. It may be interesting to those who are working in this field. However, this manuscript is with too many typos and grammar problems. In many areas, I don't what the authors try to say. I suggest that the author ask a native English speaker to edit your manuscript.

Dear Editor,
Microbiology Spectrum

Subject: Response to Reviewer's Comments

Kindly, refer to your email May 4, 2022, regarding our submitted manuscript ID (Spectrum01155-22) entitled “**Effects of milk replacer-based Lactobacillus on growth and gut development of yak's calves; A gut microbiome and metabolic base study**”. Based on reviewer comments, we, the authors, have carefully incorporated suggestion to the manuscript, and we hope that the revised-manuscript now will meet the standards for publication in your good journal.

We are looking forward to hearing from you soon.

Reviewer 1:

No.	Comments	Response
1	Overall, the manuscript is interesting, the methods are appropriate, and the results are well presented. The manuscript could use an additional spelling and grammar check, and I included some minor corrections here: line 14: correct to "highlights"line 26" remove the hyphenline 29: correct to "probiotics dietary supplements changes the"line 33: correct to "collaterally"line 39, 454, 458, 460, Fig 1 legend: correct to "yak calves"line 42: italicize the species' nameline 60: correct to "probiotics is suspected to be a vital feeding strategy as"line 85: "establishment of microbiota"line 103: "yak calves' microbiome"line 108: I would change 'microbiome' to 'bacterial	We are very thankful for your comments. We have double-checked the whole manuscript for language errors according to your suggestions. In addition, we have invited a native English speakers to review the language of the manuscript.

	community' here and elsewhere in the manuscript because 16S only gives a fraction of the microbiome and is misleading to use. line 136: correct to "diversity" lines 158 - 159: italicize the genera names here line 193, 248, 250, 253, 323: revise to "yak calves" line 404: "vitamins"	
	line 62: while this sentence is true, since this paper focuses on cattle it would be more informative to include a reference on improving calf health here instead of one on human infants.	Respected reviewer, we have updated the sentence accordingly.
	lines 68- 76: was this research on any species of Lactobacillus? I think many of the benefits are species or strain specific, so the authors should be more specific. For example, "...therapy of certain Lactobacillus species modulated..."	Respected reviewer, actually the study is not focusing any specie of Lactobacillus. We have made more specific description in manuscript according to your suggestion.
	lines 185: I am not sure what the authors means by "kept tendancy", please revise.	We have made appropriate changes for better clarity.
	Figures: the text in the figures was too small to read since there were so many panels. I recommend moving some of these to supplementary, or splitting figures into two as needed to make the panels and text larger.	We have made appropriate changes in Figures 6. Moreover, we moved some results to the supplementary section.

Reviewer 2:

No.	Comments	Response
-----	----------	----------

1	The manuscript submitted by Li et al. demonstrates a milk replacer-based lactobacillus in the yak's calves could increase the body weight. It may be interesting to those who are working in this field. However, this manuscript is with too many typos and grammar problems. In many areas, I don't what the authors try to say. I suggest that the author ask a native English speaker to edit your manuscript.	Respected reviewer, we have double-checked the whole manuscript for language errors. In addition, we have invited a native English speakers to review the language of the manuscript.
----------	--	---

Kind regards,

Prof. Jiakui Li

College of Veterinary Medicine, Huazhong
Agricultural University, Wuhan, China.

Email: lijk210@sina.com

June 8, 2022

Prof. Jiakui Li
College of Veterinary Medicine, Huazhong Agricultural University, Wuhan, 430070, PR China
Shizishan Road 1
Wuhan
China

Re: Spectrum01155-22R1-A (Effects of milk replacer-based Lactobacillus on growth and gut development of yak calves; A gut microbiome and metabolic base study)

Dear Prof. Jiakui Li:

Your manuscript has been accepted, and I am forwarding it to the ASM Journals Department for publication. You will be notified when your proofs are ready to be viewed.

Sincerely,

John Chaston
Editor, Microbiology Spectrum
